# CXCR2 Antagonist RIST4721 Acts as a Potent Chemotaxis Inhibitor of Mature Neutrophils Derived from Ex Vivo-Cultured Mouse Bone Marrow

**DOI:** 10.3390/biomedicines11020479

**Published:** 2023-02-07

**Authors:** Klaudia Szymczak, Margery G. H. Pelletier, James M. Mackay, DeAnne Reid, Peter C. W. Gaines

**Affiliations:** 1Department of Biological Sciences, Biomedical Engineering and Biotechnology Program, University of Massachusetts Lowell, Lowell, MA 01854, USA; 2Aristea Therapeutics, 12770 High Bluff Drive, #380, San Diego, CA 92130, USA

**Keywords:** neutrophil, chemotaxis, CXCR2 antagonist, inflammatory disease, phagocytosis, respiratory burst

## Abstract

Neutrophils act as critical mediators of innate immunity, which depends on their rapid responses to chemokines followed by their migration towards sites of infection during chemotaxis. Chemokine receptors expressed on the surface of neutrophils mediate chemotaxis by activating contractile machinery as the cells escape from capillary beds and then attack pathogens. Neutrophils also contribute to inflammatory responses, which support pathogen destruction but can lead to acute and chronic inflammatory disorders. CXCR2, a G-protein-coupled chemokine receptor expressed on both myeloid and epithelial cells, is well-characterized for its capacities to bind multiple chemokines, including interleukin-8 and growth-related oncogene alpha in humans or keratinocyte chemokine (KC) in mice. Here we show that a small molecule CXCR2 antagonist termed RIST4721 can effectively inhibit KC-stimulated chemotaxis by neutrophils derived from ex vivo-cultured mouse bone marrow in a potent and dose-dependent manner. Antagonistic properties of RIST4721 are thoroughly characterized, including the maximal, half-maximal and minimum concentrations required to inhibit chemotaxis. Importantly, RIST4721-treated neutrophils exhibit robust phagocytosis and reactive oxygen species production, confirming drug specificity to chemotaxis inhibition. Together our data indicate that RIST4721 acts to inhibit inflammation mediated and potentiated by neutrophils and therefore promises to facilitate treatment of a host of inflammatory conditions.

## 1. Introduction

Neutrophils play a pivotal role in innate immune responses as they exhibit rapid migration towards sites of infection or inflammation, and then activate effective defensive mechanisms leading to pathogen destruction. These professional phagocytes, also termed polymorphonuclear neutrophils (PMNs), mature from hematopoietic stem cells in the bone marrow through a sequential, cytokine-regulated differentiation process with distinct stages that include common myeloid progenitors (CMP), promyelocytes, several myelocytic stages (myelocytes, metamyelocytes, band cells) and finally mature PMNs [1,2]. This maturation process is regulated in part by the granulocyte colony-stimulating factor (G-CSF) receptor upon binding of its ligand, which activates multiple intracellular pathways that ultimately control the expression of neutrophil-specific genes [3,4]. As PMNs mature, they undergo unique morphologic changes including condensation of chromatin, unique nuclear envelope component changes during the formation of characteristic segmented or multilobed structures and decreased nuclear-to-cytoplasmic ratios (N/C) [5]. Mature neutrophils are then released from large pools in the bone marrow into circulation, where they are poised to respond to gradients of chemoattractants secreted by infected tissues in the process of chemotaxis. Chemotaxis is regulated by members of the CXC subfamily of chemokines that are characterized by the presence in their N-termini of four conserved cysteine residues, the first two of which are separated by a variable amino acid [6]. Among the 17 known CXC chemokine subfamily members is CXCL8 (also termed IL-8 and neutrophil-activating protein), which plays a critical role in neutrophil chemotaxis and pro-inflammatory responses. CXCL8 can act through either of two G-protein-coupled receptors that can effectively bind the chemoattractant, CXCR1 and CXCR2 [7,8]. Upon binding CXCL8, CXCR1/2 causes dissociation of the GTP-binding protein subunits, G-α and the G-βγ complex, which independently activate pathways mediated by phosphatidylinositol 3-kinase (PI3K), phospholipase C (PLC) and p38 mitogen-activated protein kinase (p38MAPK). Once activated, these pathways are essential for integrin upregulation and reorganization of the actin cytoskeleton during cell migration [9,10,11,12]. In mice, homologous versions of IL-8 and CXCR1 are not present; however, CXCR2-binding keratinocyte chemoattractant (KC) is considered a reliable functional homologue of human IL-8 and is commonly used in mouse studies on neutrophil migration [13].

Neutrophil functions are well established as critical to innate immunity, as evident from neutropenia that is associated with reduced life expectancy [14,15]. However, persistent infiltration of neutrophils has been linked to multiple autoimmune and inflammatory diseases, including but not limited to chronic obstructive pulmonary disorder (COPD), cystic fibrosis, asthma, rheumatoid arthritis, psoriasis, neutrophilic dermatoses and inflammatory bowel diseases [16]. Moreover, neutrophils are recruited to the tumor microenvironment, the detection of which often correlates with tumor progression and poor clinical prognosis [17,18,19,20,21]. Current treatments prescribed to ameliorate the detrimental effects of excessive neutrophil infiltration are microtubule inhibitors (MTI) such as colchicine, and substances that promote the shedding of adhesion molecules critical to neutrophil extravasation, for example the integrins LFA-1 (CD11a/CD18) and Mac-1 (CD11b/CD18) [22]. Such treatments include methotrexate, glucocorticoids (GC) and nonsteroidal anti-inflammatory drugs (NSAIDS). Although effective in some cases, these therapies are not considered specific to neutrophil chemotaxis, can lead to uncontrollable responses and present serious side effects, especially with long-term use [23,24]. Alternative approaches to address these issues include the use of ligand neutralizing antibodies, small molecule inhibitors of PI3K, PLC and p38MAPK pathways, or microRNA-mediated silencing of chemokine expression [25]. However, reversibly inhibiting chemokine receptors is arguably the most promising out of all the investigated strategies to control neutrophil chemotaxis and recruitment.

Multiple compounds that antagonize chemokine receptors have been developed, two of which have been approved by the FDA: Maraviroc, a CCR5 antagonist used for HIV-1 infection, and Plerixafor, a CXCR4 antagonist used as a hematopoietic stem cell mobilizer in patients with non-Hodgkin lymphoma and multiple myeloma [26,27]. Thus, targeting CXCR2 could be a potent method to regulate neutrophil migration as a form of therapy for a variety of inflammatory disorders. In particular, a CXCR2 antagonist could be particularly effective in treating disorders associated with abnormally high expression of IL-8 or its receptor, including psoriasis, palmoplantar pustulosis (PPP), acute respiratory distress syndrome (ARDS) and rheumatologic conditions such as familial Mediterranean fever (FMF) and Behcet’s disease [28,29,30,31,32,33,34,35]. Ligands that can be blocked by such an antagonist include the epithelial-cell-derived neutrophil-activating protein (ENA-78), growth-related oncogene alpha (GRO-α; also known as CXC chemokine ligand-1 [CXCL1]) and CXCL8, which are produced by macrophages, mast cells and epithelial cells [36,37,38,39]. In the studies presented here, we aimed to characterize the effects of a novel small molecule CXCR2 antagonist developed by Aristea Therapeutics (CA), termed RIST4721, on mouse neutrophil chemotaxis. RIST4721 was designed to reversibly block human CXCR2 via allosteric interactions, which is predicted to inhibit neutrophil chemotaxis and thereby prevent neutrophil migration to sites of inflammation. We used mouse neutrophils derived from ex vivo culture of bone marrow that was treated with a stepwise combination of the cytokines known to promote neutrophil differentiation [40,41]. Our results demonstrate that we have designed an optimal assay for testing mouse neutrophil chemotaxis and that RIST4721 effectively inhibits neutrophil chemotaxis. Data showing dose-dependent responses to RIST4721 are also presented along with results that identify the minimal, maximal and inhibitory 50 concentrations (IC50). Importantly the effects of RIST4721 are specific to chemotaxis, as additional functional responses that include phagocytosis and the production of reactive oxygen species (ROS) during the respiratory burst were not affected by this CXCR2 antagonist. These results should be informative for clinical trials of RIST4721 as an effective therapeutic to treat multiple inflammatory diseases caused by aberrantly recruited neutrophils, a double-edged sword of the innate immune system.

## 2. Materials and Methods

### 2.1. Drug Preparation

Small molecule CXCR2 antagonist RIST4721 was provided as a powder by Aristea Therapeutics (San Diego, CA, USA) and stored at −20 °C in vials protected from light. Stock solutions of the drug were prepared every 2–3 months by dissolving appropriate portions of the powder in DMSO (Thermo Fisher Scientific, Waltham, MA, USA) at 500 mM final concentration. Single-use aliquots were then prepared to avoid multiple freeze–thaw cycles. Working solutions were made if necessary by diluting the stock solution in DMSO immediately prior to use.

### 2.2. Mice and Ex Vivo Neutrophil Culture

Eight-week-old C57BL/6 female mice were purchased from Charles River Laboratories (Wilmington, MA, USA) and euthanized under protocols approved by the UMass Lowell Institutional Animal Care and Use Committee to harvest bone marrow flushed from femurs and tibias, which were frozen and stored in liquid nitrogen. Bone marrow batches were then thawed as needed and hematopoietic stem cells (HSCs) were isolated to produce mature neutrophils using previously established protocols [40,41]. Briefly, HSCs were separated through negative selection using BD IMag™ Mouse Hematopoietic Progenitor Cell Enrichment Set (BD Biosciences, Woburn, MA, USA) according to manufacturer’s recommendations. The cells were then cultured for 3 days in basal media (Iscove’s modified Dulbecco’s media (IMDM; HyClone, Logan, UT, USA) with 20% horse serum (HS; Gibco, Grand Island, NY, USA) and penicillin (50 U/mL)/streptomycin sulfate (50 μg/mL; HyClone)) supplemented with SCF (50 ng/mL; Peprotech, Rocky Hill, NJ, USA) and IL-3 (50 ng/mL; Peprotech) at 37 °C and 5% CO_2_. On day 3 (D3 CMP stage) the media was changed to basal media supplemented with SCF (50 ng/mL), IL-3 (50 ng/mL) and G-CSF (50 ng/mL; Peprotech) and the cells were cultured for 2 days at the same conditions. Finally, on day 5 (D5 pro-PMN stage) the media was changed to basal media with G-CSF (50 ng/mL) only and the cells were cultured for additional 2 days until terminal maturation (D7 PMN). The differentiation of cells was confirmed by morphologic evaluation of cells stained with Wright plus Giemsa (Sigma-Aldrich, St. Louis, MO, USA) and detection of neutrophil-specific markers as described below.

### 2.3. Chemotaxis Assays and Half-Maximal Inhibitory Calculation

Chemotaxis experiments were performed in 96-well HTS transwell plates with 3 μm pores (Corning, Corning, NY, USA) using a previously established protocol [42]. The assay settings were further optimized by testing various cell numbers (between 2 × 10^4^ and 2 × 10^5^), different concentrations of KC (10 to 250 ng/mL; Peprotech) and total migration times (30 min to 3 h) to maximize the chemotactic responses. To test the effect of RIST4721 on chemotaxis, solutions of phenol red-free IMDM plus 1% certified FBS (Gibco) with or without KC (100 ng/mL) were prepared, with DMSO used as the vehicle control. To prepare cells for each test, mature neutrophil (D7 PMN) concentrations were adjusted to 5 × 10^5^ cells/mL by diluting the cells in basal media with G-CSF only. The cells were mixed to ensure even distribution and plated in 6-well plates at 5 mL per well. The appropriate working concentration of RIST4721 (or DMSO as control) was then added into the cultures at 5 μL per well. The cells were incubated with the drug for 30 min unless otherwise indicated, 2 × 10^5^ cells per replicate were then harvested, centrifuged at 250× *g* for 5 min, washed in 5 mL of PBS and again centrifuged. Cell pellets were then resuspended in phenol red-free IMDM at 80 μL per replicate and the drug (or DMSO) was added at the same concentrations as used in the pretreatment step. Prepared media were then plated into bottom chambers of designated wells at 230 μL per well and 5 replicates per condition. Next, the insert plate with top chambers was applied. Finally, the cells were transferred into the top chamber of the chemotaxis plates, and plates were incubated at 37 °C/5% CO_2_ for 2 h, unless otherwise stated. After the incubation, the top chambers were removed and 230 μL of CellTiter-Glo (Promega, Madison, WA, USA) was added into each well according to the manufacturer’s protocol. The plate was incubated at RT for 10 min and viable cells were detected by measuring luminescence using a microplate reader (Synergy HT, Biotek, Winooski, VT, USA). The averages of luminescence signals or fold changes vs. DMSO control were calculated for each tested condition to generate dose–response graphs, and values were uploaded into an online EC50 calculator (ATT Bioquest, Pleasanton, CA, USA) to identify half-maximal inhibitory concentration (IC_50_) values.

### 2.4. Cytospins and Wright/Giemsa Staining

To visualize cellular morphology, 1 × 10^5^ cells were harvested from the culture and centrifuged at 1400× *g* for 5 min. The pellet was then resuspended in 300 μL PBS with 0.1% BSA (Thermo Fisher Scientific) and transferred into a cytospin apparatus for slide preparation. The cells were centrifuged for 5 min at 55× *g* and the slides were air-dried for 5 min followed by staining in Wright stain for 2.5 min. The slides were then submerged in Sorensen’s phosphate buffer (pH 6.8) for 5 min, stained in Giemsa stain for 15 s, and rinsed twice in distilled water for 5 min each. Finally, the specimens were air-dried and mounted with a drop of Permount (Fisher Scientific) plus a glass coverslip. The cells were then imaged under a 60X oil immersion objective using an Olympus BX41 microscope (Olympus, Waltham, MA, USA) fitted with an Olympus DP camera and Controller image analysis software (Olympus, Center Valley, PA, USA). 

### 2.5. Proliferation and Viability Assays

To assess the effect of RIST4721 treatment on proliferation and differentiation of ex vivo progenitors, D3 CMPs were plated in triplicate wells at 2 × 10^5^ cells/mL (12-well plate, 2.5 mL per well) and the drug (20 nM) or DMSO were added into the media. The cells were cultured as described above until they reached the mature neutrophil stage (D7 PMN), making sure that the drug and DMSO were replenished during media changes and passages. Cell counts were performed for each replicate at 24, 48, 72 and 96 h timepoints using trypan blue exclusion assays (HyClone) and the total cell numbers were plotted as proliferation curves. For viability assays, untreated D7 PMNs were mixed and split into triplicate wells for RIST4721 (10 μM) or DMSO treatment. The cells were incubated for 2.5 h at 37 °C and 5% CO_2_, after which 100 μL samples were collected from each well and transferred into a white, clear bottom 96-well plate. CellTiter-Glo reagent was then added into each sample at 1:1 ratio and the plate was incubated at RT for 10 min. Viable cells were detected by luminescence measurements using a microplate reader (Synergy HT). 

### 2.6. Cell Surface Marker Analysis

Cells at D3 CMP stage were treated with 20 nM RIST4721 (or DMSO) and differentiated into mature neutrophils as described above while replenishing the drug and DMSO during culture expansion or media changes. Cell surface marker analysis using fluorescent immunolabeling and imaging flow cytometry was then performed as previously described [42]. Briefly, the cells were harvested at 1 × 10^6^ per sample through centrifugation at 250× *g* for 5 min. The pellets were then washed in PBS and centrifuged again. The cells were resuspended in PBS with 2% FBS at 100 μL per sample, and Fc Block (Purified Rat Anti-Mouse CD16/CD32; BD Biosciences) was added into each sample at 50 μg/mL followed by incubation on ice for 15 min. The cells were then stained with PE-conjugated anti-Cd11b, FITC-conjugated anti-Ly6G, or corresponding isotypes (4 μg/mL for each antibody; BD Biosciences) for 45 min on ice while protected from light. Finally, the samples were washed in 400 μL of PBS with 2% FBS, centrifuged at 1400× *g* for 5 min and resuspended in 35 μL of fresh PBS with 2% FBS for processing with FlowSight imaging flow cytometer (Luminex, Austin, TX, USA).

### 2.7. Phagocytosis

Neutrophils at the D7 PMN stage were pretreated with 20 nM RIST4721 (or DMSO) by adding drug to the culture media and incubating the cells for 30 min at 37 °C, 5% CO_2_. Next, 1 × 10^6^ cells per sample were harvested by centrifugation at 250× *g* for 5 min, washed in PBS and resuspended in HBSS at 100 μL per sample. The drug and DMSO were added to the cell mixtures at the same concentration as used for the pretreatment step. The cells were then subjected to a phagocytosis assay as described previously [41]. Briefly, the cell/drug mixtures were transferred into 5 mL snap-cap tubes together with 710 μL of HBSS, 100 μL of mouse serum, 80 μL of NucBlue reagent and 10 μL of opsonized pHrodo *E. coli* bioparticles (Molecular Probes, Eugene, OR, USA). The tubes were secured with parafilm and incubated at 37 °C with gentle rocking for 1 h. The samples were then harvested through centrifugation at 1000× *g* for 5 min and resuspended in 35 μL HBSS with 2% FBS for processing with FlowSight imaging flow cytometer (Luminex). 

### 2.8. Respiratory Burst

Neutrophils (D7 PMNs) were pretreated 30 min at 37 °C, 5% CO_2_ with 20 nM RIST4721 (or DMSO with equivalent volume), 1 × 10^6^ cells were harvested by centrifugation (250× *g*, 5 min), washed in PBS and resuspended in HBSS with 0.1% glucose (Sigma-Aldrich) at 160 μL per replicate. Respiratory burst was then stimulated and amounts of ROS were measured as described previously [42]. In short, the cells were transferred into replicate wells in white, clear bottom 96-well plate at 160 μL per well. Diogenes (National Diagnostics, Atlanta, GA, USA) was then added into each sample at 40 μL per well, the plate was incubated at 37 °C for 3 min, and the cells were stimulated with either phorbol myristate acetate (PMA, 0.5 μg/mL; Sigma-Aldrich) or opsonized Zymosan (OZ, 0.5 mg/mL; Sigma-Aldrich). The luminescence signal was detected every 2 min for 2 h using a microplate reader equipped with a kinetic mode (Synergy HT).

### 2.9. Data and Statistical Analysis

All the data obtained from imaging flow cytometry analyses were processed using provided IDEAS software (Luminex) to generate histograms and representative images of stained cells. The data from microplate-reader-based assays were analyzed in Excel software (Microsoft Corporation, Redmond, WA, USA) to calculate averages ± SD and *p*-values using unpaired Student *t*-tests assuming equal variances. All the presented results were produced using at least three technical or biological replicates and are representative of at least three independent assays, unless otherwise stated.

## 3. Results

### 3.1. Bone-Marrow-Derived Mouse Neutrophils Exhibit Typical PMN Characteristics Prior to Chemotaxis Assays

Neutrophils used for all chemotaxis tests were first assessed for their hallmark lobulated nuclei by visual inspection of Wright–Giemsa-stained cells after cytocentrifugation. As depicted in Figure 1a, the expanded CMP population after 3 days of culture exhibited round or ovoid nuclear morphologies with high N/C ratios, whereas the vast majority of D7 PMN cells showed lobulated nuclei with low N/C ratios. The D3 CMP and D7 PMN cells then were immunolabelled with fluorescent antibodies targeting the mature neutrophil markers Ly6G and Mac-1, and then expression patterns were analyzed using imaging flow cytometry. As shown in Figure 1b,c, D3 CMPs exhibit little to no expression of either cell surface marker, whereas both cell surface markers are readily detected in D7 PMNs, validating our neutrophil model for in-depth chemotaxis assays. 

### 3.2. Quantitative Chemotaxis Measurements Depend on Cell Numbers, Chemokine Concentration and Migration Time

To ensure optimal and reproducible results for our planned tests of chemotaxis inhibition by RIST4721, we used our standardized, transwell-based migration assay [42] but varied several parameters including cell numbers used in each assay, concentrations of KC and total migration times prior to cell number quantifications with CellTiter-Glo. We began by confirming the correlation of neutrophil numbers to luminescence generated by the CellTiter-Glo reagent using two different ranges of cell numbers (10,000–50,000 or 20,000–100,000). As shown in Appendix A, R2 values were in the linear range for both tests (0.972 and 0.996, respectively), demonstrating that luminescence detected in the bottom chamber of the chemotaxis devices should accurately reflect the number of migrated cells, at least up to 1 × 10^5^ neutrophils. Chemotaxis of different numbers of cells added to the chambers was next analyzed; as shown in Figure 2a, the ex vivo culture-derived neutrophils exhibited robust chemotactic activity when responding to KC (100 ng/mL) as compared to FBS alone, but the responses increased proportionally to the numbers of cells used in each test. The highest migration levels, and perhaps more importantly the best fold increases between KC-stimulated cells vs. FBS alone, were observed for 2 × 10^5^ cells (almost 26,000 RLU and 7.5-fold increased response, respectively); thus, this total cell number was chosen for all the subsequent assays. Higher numbers of cells were not considered due to the risk of membrane overcrowding and numbers exceeding the linear range of luminescence provided by the CellTiter-Glo reagent (e.g., no more than 1 × 10^5^ total migrated cells). Next, we tested a wide range of KC concentrations (10–250 ng/mL) and also observed an increasing trend in chemotaxis levels, but concentrations higher than 100 ng/mL showed little change, and therefore this dose was considered optimal (Figure 2b). Finally, we investigated various migration times to identify that required to reach maximal numbers of neutrophils in the bottom chamber; since no more were gained after 2 h of incubation, this time was chosen for all subsequent studies (Figure 2c).

### 3.3. RIST4721 Inhibits KC-Mediated Chemotaxis in a Dose-Dependent Manner

RIST4721 was designed as a small molecule antagonist of CXCR2 that can be provided as an oral treatment but here was dissolved in DMSO for use in the chemotaxis media (see Figure 3a for chemical structure). To initially assess the inhibitory properties of RIST4721 on chemotaxis, we treated the ex vivo bone marrow-derived neutrophils with various concentrations of the drug ranging from 1 nM to 100 μM to identify both minimal and maximal levels of chemotaxis inhibition, using DMSO at the maximal volume of added drug as the negative control. The parameters for each assay were based on the optimization analyses with KC, specifically 2 × 10^5^ cells total, 100 ng/mL of KC and 2 h of incubation per test, each used prior to quantifying migrated cell numbers with CellTiter-Glo. Experimental conditions included a 30 min pre-incubation with RIST4721 or DMSO. We began with the lower end of concentrations, demonstrating that 1 nM RIST4721 showed little inhibition (the *p*-value between this and no drug was 0.24), but impressive results were observed with 10 nM and 100 nM (Figure 3b). Responses above 100 nM were then examined, which showed that 1 μM RIST4721 completely abrogated any additional chemotaxis stimulated with KC (e.g., above that caused by FBS alone), indicating the complete inhibition of CXCR2 activities (Figure 3c). RIST4721 did not change FBS-mediated responses, indicating chemotaxis inhibition is CXCR2-specific. Moreover, chemotaxis inhibition by RIST4721 is rapid as pre-incubation times longer than 30 min did not enhance the inhibitory efficacy of the drug at 10 nM concentrations (Figure 3d).

Given the range of decreased chemotaxis observed by neutrophils treated with 10 nM vs. 100 nM concentrations of RIST4721 (specifically 50% vs. 70% decreased chemotaxis compared to no drug, respectively), we wished to dig deeper into the concentration-dependent effects of RIST4721 in order to identify the minimal inhibitory concentration (MIC) required to cause statistically significant changes in neutrophil migration. Assays with smaller increments in drug concentrations were therefore performed, which showed that either 1 nM or 5 nM of the drug could cause a small decrease in chemotaxis, but the results were inconsistent as revealed by statistical assessment of the differences in chemotaxis vs. DMSO (Figure 3e and Appendix A). However, consistent and substantial chemotaxis inhibition was observed with 10 nM RIST4721, with a 36% vs. 26% reduction in KC-stimulated migration vs. the DMSO (Figure 3e). This further testing at fine-tuned ranges of drug concentrations suggest that the MIC is around 5 nM. Importantly, these data showed that the IC_50_ (the concentration required for half-maximal inhibition) is most likely near 20 nM (responses shown in Figure 3e were 54% lower than those generated by DMSO). As shown previously, treatment with the drug did not affect neutrophil migration in wells with serum alone, confirming that the observed inhibitory effects are specific to CXCR2-mediated pathways.

### 3.4. RIST4721 Is a Potent Inhibitor of Neutrophil Chemotaxis with Moderately Low IC50

Based on the trends revealed from the chemotaxis assays with RIST4721 in Figure 3, we performed further tests to accurately calculate the drug’s IC_50_, as this is a useful FDA-recommended measurement that can help predict in vivo potency of the drug. We first analyzed chemotaxis values from several independent assays using RIST4721 concentrations ranging between 5 nM and 10 μM, as an example, input the values into an IC50 calculator available through AAT Bioquest and found that the IC_50_ values oscillated between 14 nM and 28 nM (representative data are shown in Figure 4a). We next converted the RLU values from these multiple chemotaxis assays into fold changes vs. the DMSO control, calculated the average fold change for specific doses and then used the IC50 calculator to establish the final IC_50_ value for RIST4721 at ~17 nM (Figure 4b).

### 3.5. Treatment with RIST4721 Does Not Affect Progenitor Differentiation Nor Viability

To rule out the possibility that the lower numbers of migrating cells in RIST4721-treated cultures is caused by the impaired survival or maturation of the tested neutrophils, we performed a thorough analysis of their viability and differentiation characteristics with drug treatment. First, we investigated whether the presence of RIST4721 (20 nM) changes the growth profile of differentiating neutrophils starting from the common myeloid progenitor (CMP) stage through to the PMN stage. Our results show no significant differences between the untreated, DMSO-treated and drug-treated cultures, demonstrating that RIST4721 does not alter the proliferation rates of mitotically active neutrophil progenitors (Figure 5a). Second, we used the CellTiter-Glo assay to assess the viability of PMNs when exposed to the maximum inhibitory concentration of RIST4721 (10 μM) and again observed no notable differences between the drug-treated cells and controls (Figure 5b). 

As mentioned previously, nuclear lobulation and increased expression of two cell surface proteins, Mac-1 and Ly6G, serve as hallmarks of neutrophil differentiation with our model cells. Consistent with these phenotypic changes, significant nuclear lobulation was observed in treated neutrophils that were indistinguishable from untreated or DMSO-treated cells, as shown in Figure 5c. Moreover, flow cytometry analyses of cell surface markers revealed almost identical levels of Mac-1 and Ly6G expression in samples from drug-treated vs. control cultures (Figure 5d). Together, these data indicate that RIST4721 does not disrupt myeloid progenitor growth nor does it affect their commitment to the neutrophil lineage, and therefore will not alter basic neutrophil differentiation characteristics.

### 3.6. Neutrophils Treated with RIST4721 Show Normal Phagocytosis and Respiratory Burst Responses

In order to ensure RIST4721 does not disrupt molecular pathways associated with other innate immune responses provided by neutrophils, we subjected the drug-treated cells to phagocytosis and respiratory burst assays. For the former, we used pHrodo FITC-conjugated *E. coli* bioparticles to imitate pathogen exposure and analyzed the treated cells for phagocytosis percentages with imaging flow cytometry. As shown in Figure 6a, the distributions of pHrodo positive cells were very similar between RIST4721-treated cells and the controls, with 57.8% phagocytosing cells in the drug-treated sample vs. 62.2% and 63.3% in DMSO-treated and untreated samples, respectively (the negative control of undifferentiated cells essentially lack engulfed particles, see [41]). To quantify the respiratory burst, we stimulated the cells with two potent inducers of ROS production, PMA and OZ, and detected the released oxygen radicals (primarily superoxide anion, see [43]) in real time using an enhanced luminol reagent (Diogenes) and a microplate reader. Again, the results indicate no distinguishable difference between RIST4721-treated cells vs. untreated or DMSO-treated cells, with robust amounts of ROS detected under all conditions (Figure 6b). Collectively, our data provide important evidence that RIST4721 will not interfere with pathways that promote phagocytosis or the respiratory burst, both critical to the capacity of neutrophils to attack and kill pathogens in an infected patient. The combined data therefore indicate that RIST4721 acts as a potent and specific inhibitor of CXCR2, and treatments with RIST4721 promise to suppress aberrant neutrophil migration but not block other functional responses, including their capacities to engulf pathogens and produce antimicrobial weapons that include ROS.

## 4. Discussion

Excessive accumulation of neutrophils in tissues is associated with numerous inflammatory diseases as the aberrantly recruited leukocytes cause cellular damage through the degranulation and production of toxic reactive oxygen radicals. Neutrophils also contribute to inflammatory conditions with their capacity to release neutrophil extracellular traps (NETs), a means of trapping extracellular bacteria with a net of ejected DNA decorated with proteolytic enzymes. These antimicrobial functions not only promote inflammation but also destroy diseased tissues. An important example is chronic obstructive pulmonary disease (COPD), in which neutrophils colonize the airways and release high amounts of proteases including neutrophil elastase (NE) and matrix metalloproteinase, which then digest extracellular matrix proteins including collagen, elastin and fibronectin, thereby causing alveolar destruction, mucous secretion and airway obstruction [44,45]. Similar neutrophil activity is associated with other pulmonary conditions such as asthma, bronchiectasis and cystic fibrosis, each often associated with NE release (recently reviewed in [46]). Even viral infections that target the pulmonary system involve the recruitment of neutrophils, exemplified by the relatively recent discovery that SARS-CoV-2 infection, the causative agent of Coronavirus disease 19, causes the release of NETs that not only increase lung inflammation but can also promote the cytokine storm often associated with this disease [47]. Enhanced neutrophil infiltration and function is also a key factor in the pathogenesis of skin disorders collectively called neutrophilic dermatoses (NDs). This group encompasses a growing number of cutaneous conditions including palmoplantar pustulosis (PPP), pyoderma gangrenosum (PG), Sweet’s syndrome (SS), subcorneal pustular dermatosis and generalized pustular psoriasis [48]. Interestingly, patients with ND exhibit increased levels of serum G-CSF that supports increased neutrophil production, along with elevated IL-17 levels in the skin, a cytokine that promotes neutrophils to produce IL-8 and thereby causes increased recruitment and pro-inflammatory responses in a vicious feedback pathway [49]. Combining these results provides strong support for the use of chemokine receptor antagonists that will target neutrophil chemotaxis, thereby suppressing their roles in these types of lung and skin disorders. Our data indicate that such suppression is possible with nanomolar concentrations of RIST4721, but that the level of suppression provided by the drug will not affect neutrophil maturation nor other functions critical to innate immunity. RIST4721 may also be used as an adjunct for other autoimmune-mediated inflammatory conditions or lung inflammation caused by respiratory-targeting viruses including COVID-19.

Recent studies also reveal a clear involvement of neutrophil recruitment in tumor development and progression, potentially caused by the immunosuppression of T cells and promoting angiogenesis and metastasis (reviewed in [50]). Indeed, the depletion of neutrophils in several breast and ovarian cancer models resulted in reduced tumor growth and density [51,52,53]. Moreover, Houghton et al. recently reported that neutrophil elastase activates pathways stimulating growth and angiogenesis in a murine model of lung adenocarcinoma [54]. Finally, it has been shown that neutrophils can interact with circulating cancer cells via Cd11b- or NETs-mediated mechanisms and facilitate their retention in metastatic tissues including the lungs and liver [55,56]. RIST4721 might therefore be a useful adjuvant with antitumor treatments that will suppress the recruitment of neutrophils in the tumor microenvironment. 

Studies of other small molecule antagonists of CXCR2 support the notion that RIST4721 should be further investigated in vivo for its capacity to suppress neutrophil migration in peripheral blood. For example, the CXCR2 antagonist AZD5069 (developed by AstraZeneca) is a potent compound shown to effectively block ligand binding to CXCR2 and significantly reduce chemotaxis by human peripheral blood PMNs, even at single-digit nanomolar concentrations [57]. In clinical trials, treatment with AZD5069 appeared promising as it considerably reduced neutrophil counts in sputum and lung tissue of patients with bronchiectasis and asthma; however, the drug had no effect on the frequencies of symptoms in long-term clinical trials (NCT01704495, NCT01255592). Several other compounds including SB-656933 and SCH527123 (MK-7123) presented efficacy in clinical trials when tested in patients with COPD and asthma, but markedly lower neutrophil counts observed in the peripheral blood of healthy subjects resulted in termination of the studies [58,59]. In contrast, our data indicate that RIST4721 does not affect neutrophil development (Figure 2), nor does is suppress innate immune functions other than chemotaxis (Figure 6). 

There are some limitations of the results presented here that should be considered for future studies of RIST4721 that go beyond our use of a mouse model, albeit an ex vivo one that is derived directly from bone marrow hematopoietic stem cells. One important consideration is the effective dose required to inhibit neutrophil chemotaxis once they are released into peripheral blood. Importantly, our extensive analyses using a wide range of RIST4721 indicate that concentrations as low as 1 nM suppressed chemotaxis, suggesting that low oral doses will be sufficient to achieve nM concentrations in neutrophils in serum or in target tissues (Figure 3; several tests showed statistically significant differences at 1 nM). Our results therefore suggest that RIST4721 will suppress neutrophil chemotaxis for lung or skin disorders as was previously shown for AZD5069 [60,61,62]. In addition, further cytotoxicity studies will be required with dose- and time-dependent studies using in vitro assays or ideally in vivo, despite the low toxicity observed in our tests with concentrations that suppress chemotaxis (see Figure 5). Despite these limitations, chemotaxis suppression by RIST4721 intensified proportionally to increased doses of the drug, indicating that the drug exhibits a gradual, dose-dependent effect in humans that can be fine-tuned when used as an adjuvant, in particular for the suppression of neutrophil recruitment into the tumor microenvironment. Our results also indicate that RIST4721 acts rapidly (e.g., within minutes) as pre-incubation times longer than 30 min did not significantly enhance its potency (Figure 3b). Moreover, our functional studies on RIST4721-treated neutrophils provide important evidence that this antagonist does not interfere with pathways controlling phagocytosis or respiratory burst responses, which are pivotal elements of sterile and nonsterile inflammation processes. These results are in agreement with those shown for AZD5069, indicating that these types of CXCR2 antagonists will still allow for the normal development and mobilization of neutrophils from bone marrow, and not interfere with their capacity to attack and destroy opportunistic pathogens [63,64].

## 5. Conclusions

Our results show that RIST4721 is a potent inhibitor of neutrophil chemotaxis that offers fine-tuned control of chemotaxis inhibition, with dose-dependent and highly reproducible effects. Moreover, treatment with RIST4721 does not affect the proliferation or differentiation of neutrophil progenitors, nor does it interfere with the functional properties of mature neutrophils including their capacities to phagocytose pathogens and undergo a robust respiratory burst. Continued analyses of RIST4721 for the treatment of human inflammatory conditions and/or diseases are warranted, and possible supplemental use with anticancer treatments in the tumor microenvironment should be studied.

## Figures and Tables

**Figure 1 biomedicines-11-00479-f001:**
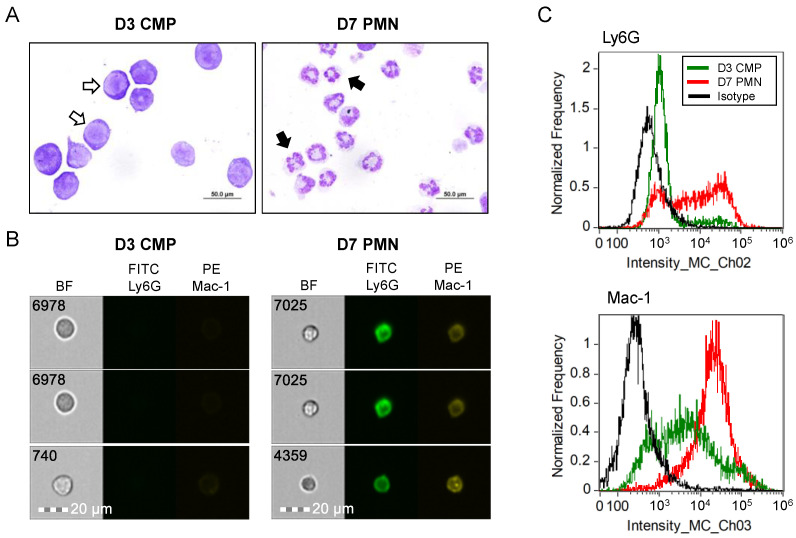
Maturation characteristics of neutrophils derived from mouse bone marrow for chemotaxis assays. (**A**) Shown are images of Wright–Giemsa-stained cells generated from ex vivo culture of bone marrow at the common myeloid progenitor stage (after 3 days of culture, D3 CMP), or the terminally differentiated stage (polymorphonuclear neutrophils after 7 days of culture, D7 PMN). CMP exhibit characteristic high N/C ratios (open arrows), whereas PMN have lobulated or ring-shaped nuclei (closed arrows). (**B**) Images are depicted of CMP and PMN labeled with anti-Ly6G or anti-Mac-1 antibodies with fluorescence tags (FITC or PE, respectively), each obtained using imaging flow cytometry. (**C**) Graphs generated from imaging flow cytometry indicate quantities of cells with either Ly6G or Mac-1 expression at each stage of differentiation, with the appropriate isotype used as the negative control.

**Figure 2 biomedicines-11-00479-f002:**
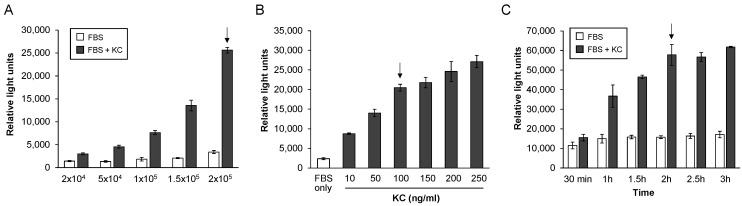
Chemotaxis responses by bone-marrow-derived neutrophils to keratinocyte chemokine (KC). Depicted are responses of neutrophils to different conditions that stimulate chemotaxis in response to either FBS alone or FBS plus added KC, each quantified using CellTiter-Glo to detect cells that have migrated into the bottom chambers of transwell plates. Shown are chemotaxis responses to (**A**) 100 ng/mL of KC with different numbers of cells added to the chambers after 2 h of incubation, (**B**) increasing concentrations of KC with 2 × 10^5^ cells total and 2 h of incubation or (**C**) different times of incubation (hours, h) after adding 2 × 10^5^ cells and 100 ng/mL of KC to the chambers. The arrow in each graph indicates optimal responses for conditions used in subsequent assays. All data shown are averages ± standard deviations (SD) from triplicate tests per condition, and represent one of at least three separate replicate assays including independent preparations of neutrophils from bone marrow.

**Figure 3 biomedicines-11-00479-f003:**
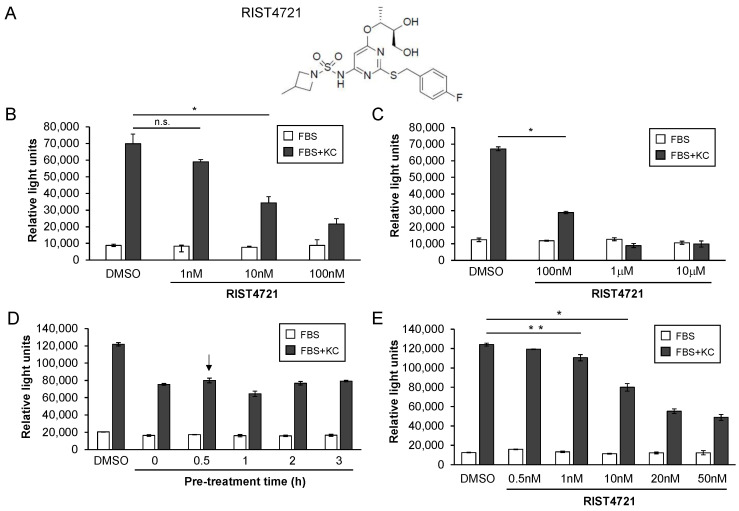
Inhibition of chemotaxis by RIST4721 at varying concentrations. (**A**) Depicted is the chemical structure of RIST4721, which is dissolved in DMSO and added to the cells both prior to chemotaxis tests and within the bottom chamber of the chemotaxis assay transwell plates. (**B**,**C**) Graphed are chemotaxis responses with neutrophils treated with low concentrations of RIST4721 (1 nM–100 nM) vs. high concentrations of drug (100 nM–10 μM). (**D**) Shown are graphed chemotaxis responses of neutrophils with different pretreatments with RIST4721, indicating a 30 min pre-incubation (arrow) is sufficient to achieve consistent chemotaxis inhibition. (**E**) Graphed are chemotaxis responses of neutrophils treated with low-range concentrations of RIST4721 used to guide further tests for MIC and IC_50_ calculations. The data shown indicate that the MIC needed to inhibit neutrophil chemotaxis oscillated between 1 nM and 5 nM. All data shown are averages ± SD from triplicate tests per assay and represent at least three independent replicates. * *p* < 0.001, ** *p* < 0.05, n.s., not significant.

**Figure 4 biomedicines-11-00479-f004:**
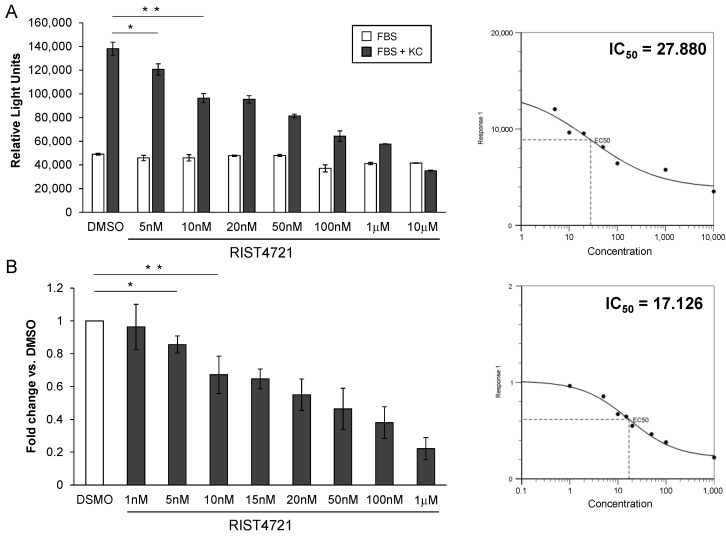
Broad-range concentration effects of RIST4721 on neutrophil chemotaxis identifies IC50 values. (**A**) Graphed are chemotaxis responses of neutrophils to a broad spectrum of RIST4721 concentrations ranging from 5 nM to 10 μM, which reveals MIC is near 5 nM but maximal inhibition (i.e., responses are equivalent to FBS alone) is provided by 10 μM; these values were then used to calculate the indicated IC_50_ (right panel). (**B**) Compiled fold changes of responses compared to DMSO are shown, which were then used to also calculate an IC_50_ value, identified as ~17 nM. Data shown are averages ± SD from triplicate tests per assay. * *p* < 0.05, ** *p* < 0.001.

**Figure 5 biomedicines-11-00479-f005:**
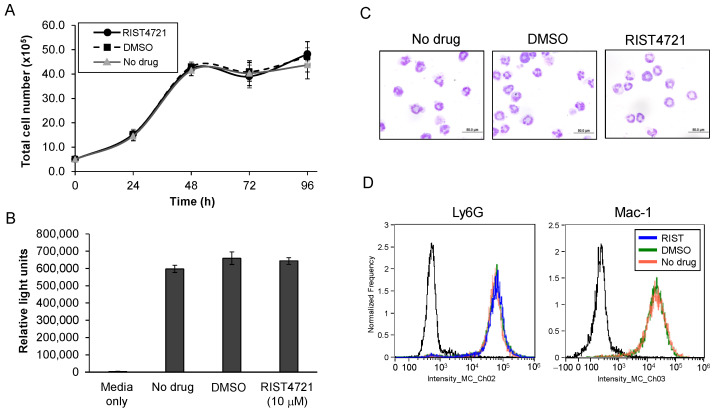
Proliferation and differentiation characteristics of neutrophils treated with RIST4721. (**A**) Graphed are total numbers of live cells identified by trypan blue exclusion in cultures treated with RIST4721 (10 nM) vs. two control conditions, DMSO (diluent) or no treatment (no drug). (**B**) Shown are levels of luminescence via CellTiter-Glo generated from mature neutrophils treated for 2.5 h with the concentration of RIST4721, shown to completely abrogate KC-mediated chemotaxis (10 μM), vs. no drug or DMSO-only treatment. Luminescence levels from media plus CellTiter-Glo with no cells (media only) are also shown to indicate background. (**C**) Pictures of neutrophils derived from cultures with RIST4721 (10 μM) vs. no drug or DMSO, each stained with Wright–Giemsa, indicate normal morphologic maturation with characteristic lobulated nuclei. (**D**) Graphed are results from flow cytometry analyses of neutrophils that were generated with RIST4721 treatment (20 nM RIST4721) vs. DMSO or no treatment, each labeled with fluorescence-labelled antibodies to either Ly6G or Mac-1. All graphed data are averages ± SD from triplicate tests per assay and represent data from three independent replicates.

**Figure 6 biomedicines-11-00479-f006:**
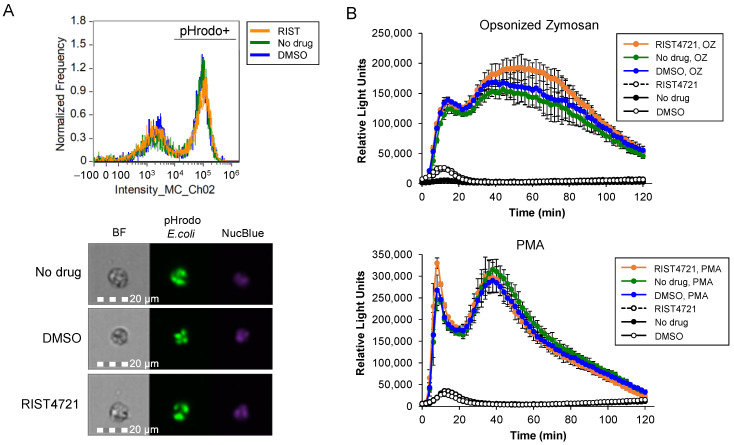
Phagocytosis and ROS production of neutrophils after treatment with RIST4721. (**A**) Shown are quantitative measurements of neutrophils that have phagocytosed *E. coli* particles that emit fluorescence when exposed to the acidic environment of the phagosome (upper panel), or images of cells with engulfed particles (lower panels), each obtained with imaging flow cytometry. (**B**) Graphed are data from analyses of ROS production by neutrophils that are treated with RIST4721 vs. no drug or DMSO, stimulated with either opsonized zymosan (upper graph) or PMA (lower graph), and measured using the enhanced luminol Diogenes. Data shown are averages ± SD from triplicate tests.

## Data Availability

Not applicable.

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
