# Peer review of "CXCR2 Antagonist RIST4721 Acts as a Potent Chemotaxis Inhibitor of Mature Neutrophils Derived from Ex Vivo-Cultured Mouse Bone Marrow"

_biomedicines, 2023, doi:10.3390/biomedicines11020479_

Round 1

Reviewer 1 Report

In the manuscript entitled “CXCR2 antagonist RIST4721 acts as a potent chemotaxis inhibitor of mature neutrophils derived from ex vivo-cultured mouse bone marrow”, Szymczak et al. investigated chemotaxis inhibitory role of CXCR2 antagonist RIST4721 in neutrophils. Treatment of RIST4721 can effectively inhibit keratinocyte chemokine-stimulated chemotaxis in a dose-dependent manner. Furthermore, RIST4721 does not reduce phagocytosis and reactive oxygen species production, suggesting drug specificity to chemotaxis inhibition. Generally, the entire work is well designed and convincing. The conclusion is supported by the experiments. My only concern is the cytotoxicity of RIST4721. It would be better to evaluate the cytotoxicity of RIST4721 in a dose- and time-dependent manner.

Author Response

Reply: We thank the reviewer for the positive comments, and agree that further cytotoxicity studies are warranted, both through more in vitro tests but also in vivo analyses. We have now added a comment at the end of the Discussion section to include such studies in our future analyses of the antagonist (see lines 492-494).

Reviewer 2 Report

Dear Authors,

Commonly well written manuscript about an interesting experimental issue. However, I would like ask you to improve the following:

1) Introduction. Please, make it shorter and clearly clarify the aim at the end of Introduction (the removed parts you can add for the Discussion, but during the reading now the reader "become lost" in the information without reaching the end...

2) Results. Fig. 6c) requests also scale bars for the other two micrographs, not only for the first one;

3) Discussion. Please, remove all references to the Figs., this do not fit here, as a discussed result should be told very shortly before the same discussion. Its not good to repeat a summary of results with a Figs.!

4) add, please, the Limitation paragraph of the provided work at the end of Discussion;

5) Conclusions. Please, shorten them and make more precise. Actually, what you need to do is - to remove the first and last sentences of the present conclusion for the end of Discussion before the last Limitation paragraph, they will fit there very well...

6) References. Could be more, additionally one is from the previous century, but I do not insist to change/remove/add something here.

After these minor objections I will recommend to publish your manuscript. Good job, thank you!

Author Response

Dear Authors,

Commonly well written manuscript about an interesting experimental issue. However, I would like ask you to improve the following:

We thank the reviewer for this positive comment, and provide our responses after each additional comment below.

1) Introduction. Please, make it shorter and clearly clarify the aim at the end of Introduction (the removed parts you can add for the Discussion, but during the reading now the reader "become lost" in the information without reaching the end...

Reply: We agree with the reviewer that the Introduction was too long and should focus primarily on the mechanisms controlling chemotaxis via CXCR2, the target of RIST4721. We have therefore made extensive edits to the Introduction to reduce it from 4 paragraphs to 3 and almost a 40% reduction in words (911 vs. 1448). We also adjusted the text in the last paragraph to specifically state the aim of the study.

2) Results. Fig. 6c) requests also scale bars for the other two micrographs, not only for the first one.

Reply: We agree that all micrographs should show the scale bars, and have now adjusted the images to include these in each figure.

3) Discussion. Please, remove all references to the Figs., this do not fit here, as a discussed result should be told very shortly before the same discussion. Its not good to repeat a summary of results with a Figs.!

Reply: We apologize for repeating summaries of the figures, and have now only included a reference to a figure if it is needed to support the discussion statements. We hope the few times that these are included is acceptable to the reviewer, but welcome any suggestions for further edits.

4) add, please, the Limitation paragraph of the provided work at the end of Discussion;

Reply: We thank the reviewer for this suggestion and appreciate the importance of adding comments about the studies’ limitations, which we now include in the edited Discussion section (see lines 480-502). We note that Biomedicines does not request a separate “Limitations” section, but appreciate that this should be addressed in the Discussion.

5) Conclusions. Please, shorten them and make more precise. Actually, what you need to do is - to remove the first and last sentences of the present conclusion for the end of Discussion before the last Limitation paragraph, they will fit there very well...

Reply: We thank the reviewer for this comment and agree that the Conclusions could be more concise. We have therefore shortened this section to briefly provide our conclusions, and have added limitations of the study to the ending paragraph of the Discussion.

6) References. Could be more, additionally one is from the previous century, but I do not insist to change/remove/add something here.

Reply: We thank the reviewer for this comment, and have removed or replaced the two previous references from the 1990’s. We also agree that certain sections could use more references, in particular those reviewing human disorders associated with increased expression of IL-8 or its receptor, which might be treated with RIST4721. Among these added references (see references 28-35 and lines 86-89) are some that date back to 1993, but more recent articles are not available and hope the reviewer agrees that these are important to include.

Reviewer 3 Report

The inhibition of chemotaxis response by RIST4721, a specific antagonist to CXCR2, was investigated on mouse bone marrow neutrophils.

Introduction is too long and should be shortened to issues concerning chemotaxis.

In description of data presented in Fig. 2, it remains unknown which KC concentration and which processing time were used in Fig. 2A.

Figs. 3-5: All figures show the concentration-dependent effect of RIST4721 on chemotaxis. These data should be combined in one representative figure. Multiple presentation of data should be avoided.

In Fig. 7, no controls are given for responses in the absence of pHrodo+, zymosan, and PMA.

The discussion about the application of RIST2741 is weak and concerns mostly the summary of main results given in section 3.

Line 458: Authors state that the luminol reagent detects released oxygen radicals. What is with the non-radical species hydrogen peroxide and hypochlorous acid?

Author Response

The inhibition of chemotaxis response by RIST4721, a specific antagonist to CXCR2, was investigated on mouse bone marrow neutrophils.

Introduction is too long and should be shortened to issues concerning chemotaxis.

Reply: We agree with the reviewer that the Introduction was too long and should focus more on mechanisms controlling neutrophil chemotaxis and disorders that might be treated with CXCR2 antagonists, and therefore have shortened this section to three paragraphs with an overall ~40% reduction in text.

In description of data presented in Fig. 2, it remains unknown which KC concentration and which processing time were used in Fig. 2A.

Reply: We apologize for not including the conditions used to test chemotaxis in Fig 2A, and have not only added the KC concentration and processing time to the Fig. 2 legend but also have added other conditions not previously mentioned to the legends for Figs. 2B and C (see new lines 290-294).

Figs. 3-5: All figures show the concentration-dependent effect of RIST4721 on chemotaxis. These data should be combined in one representative figure. Multiple presentation of data should be avoided.

Reply: We thank the reviewer for this suggestion and agree that the data in Fig. 4 should be combined with Fig. 3. We have therefore generated a new Fig. 3 with the added panel (Fig. 3E), which is sufficient for showing the minimal inhibitor concentration of RIST4721 to see statistically meaningful suppression of chemotaxis. However, we strongly feel that the data shown in Fig 4 is important to include as this provided two different IC50 concentrations, depending on the two different spectrums of concentrations used, and importantly how the values compare from biologic replicates (i.e., separate inductions and chemotaxis assays, each performed in triplicate) and combining different sets of assays in Fig. 4B as fold changes. As levels of RLU vary greatly between different biologic (i.e., separate induction) replicates in our chemotaxis assays (which include triplicate tests), we cannot combine primary data (relative light units) into one graph unless all replicates use the same concentrations (which we were able to show for Figure 4B). The separate graphs allowed us to test different combinations of concentration, which we feel are important to show.

In Fig. 7, no controls are given for responses in the absence of pHrodo+, zymosan, and PMA.

Reply: We apologize for not providing the negative controls from our ROS assays and thank the reviewer for pointing out the deficiency. We have now modified the figure, now Fig. 6, to include the responses in the absence of either OZ (Fig. 6B) or PMA (Fig. 6C) but in the presence of RIST4721 or vehicle (DMSO). For the phagocytosis data, we have previously shown that immature cells fail to engulf particles and have now stated this observation plus refer to our previous methodology manuscript that details the assay and documents this data (see reference 41 and lines 406-409).

The discussion about the application of RIST2741 is weak and concerns mostly the summary of main results given in section 3.

Reply: We agree with the reviewer that Discussion should include comments about the applications of RIST4721, and perhaps just as importantly, the limitations of our study as mentioned by reviewer #2. We have therefore reorganized the Discussion to focus on the disorders that might be treated with RIST4721 (1st and 2nd paragraphs) and include limitations of the study (last paragraph). We have also added more references regarding possible targets of RIST4721 in the Introduction (see lines 86-88), and how future studies might be similar to those performed with AZD5069 (see lines 490-505), We hope these changes, and keeping in mind the length of the Discussion, address this concern.

Line 458: Authors state that the luminol reagent detects released oxygen radicals. What is with the non-radical species hydrogen peroxide and hypochlorous acid?

Reply: We thank the reviewer for pointing out this issue regarding our assays to detect ROS with an enhance luminol reagent. We have added a comment to the line that mentions what is detected by Diogenes, which is similar to that used by Bedouhene et al to address this issue in which their studies showed that luminol primarily detects superoxide anion and only detects hydrogen peroxide in the presence of peroxidases. Although luminol can detect both intracellular and extracellular superoxide anion, most of the detected oxygen radicals is most likely extracellular (only a small portion of the luminescence was inhibited by superoxide dismutase added extracellularly), therefore the contributions by hydrogen peroxide or presumably hypochlorous acid are limited. We hope that the added comment and addition of the reference (see lines 409-412 and reference 43 in the revised manuscript) addresses this question.

Round 2

Reviewer 3 Report

Authors improved their manuscript. I have no further comments.